# Infant and young child feeding practices among mothers in the pilot Micronutrient Powder Initiative in four geographically and ethnically diverse districts in Ghana

Frank Kyei-Arthur[1], Jevaise Aballo[2‡], Abraham B. Mahama[2‡], Seth Adu-Afarwuah[3]*

1 Department of Environment and Public Health, University of Environment and Sustainable Development, Somanya, Ghana, 2 UNICEF Ghana, Accra-North, Ghana, 3 Department of Nutrition and Food Science, University of Ghana, Legon, Accra, Ghana

☯ These authors contributed equally to this work.
‡ JA and ABM also contributed equally to this work.
* sadu-afarwuah@ug.edu.gh

## Abstract

In Ghana, breastfeeding and complementary feeding counselling have been used as a nutritional intervention to promote optimal Infant and Young Child Feeding (IYCF) and nutrition. This study examined IYCF practices in four geographically and ethnically diverse districts (Ho West, Tain, Talensi and Tolon). A qualitative study involving key informant interviews (KIIs) and focus group discussions (FGDs) was conducted between November and December 2019 among women who participated in a pilot micronutrient powder intervention for children 6–23 months of age. The KIIs and FGDs were audio-taped, transcribed verbatim, and analyzed thematically using NVivo 10. Three themes emerging from the KIIs and FGDs were: level of adherence to IYCF recommendations among mothers and caregivers; IYCF recommendations perceived as the hardest to follow; and perceived motivators, facilitators, and barriers to IYCF practices. Mothers in the four districts generally followed the eight IYCF recommendations. Mothers in the Tolon district demonstrated adherence to IYCF practices, often citing the need for early initiation of breastfeeding, timely introduction of complementary feeding, and feeding children aged 9–23 months 3 times daily in addition to breastfeeding. In contrast, mothers in other districts faced challenges that hindered adherence. Giving children 4 or more food groups and timely introduction of complementary feeding were perceived as the hardest practices to follow. The main facilitators of IYCF practices include midwives and frontline nurses teaching mothers how to breastfeed, and midwives ensuring mothers initiate breastfeeding immediately after delivery. The main barriers to IYCF practices identified were insufficient breastmilk; mothers-in-law giving water to children before six months; resumption of work; and lack of financial means. Mothers in the Ho West district reported more barriers to IYCF practices, followed by mothers in the Tain, Talensi, and Tolon districts. Health practitioners, stakeholders, and policymakers should design targeted interventions that address the contextual barriers to improve IYCF practices in the various districts.

**Data Availability Statement:** All relevant data are within the paper and its Supporting Information files.

**Funding:** This study was supported by UNICEF with funding from the Government of The Netherlands. UNICEF and funders had no role in study design, data collection and analysis, decision to publish, or preparation of the manuscript.

**Competing interests:** The first author is an academic editor for PLOS ONE. This does not alter our adherence to PLOS ONE policies on sharing data and materials.

## Introduction

The World Health Organization (WHO) recommends optimal Infant and Young Child Feeding (IYCF) practices, defined as early initiation of breastfeeding within one hour after birth, exclusive breastfeeding from birth to 6 months of age, and introduction of nutritionally adequate and safe complementary (solid) foods at 6 months together with continued breastfeeding up to 2 years of age or beyond [1]. Other recommendations include: giving infants at least 5 food groups/d; feeding infants 2–3 times/d at 6–8 months of age and increasing the feeding frequency to 3–4 times/d at 9–23 months, with nutritious snacks offered 1–2 times/d as desired; ensuring food is prepared safely and hygienically; and feeding in response to infants' cues [2]. Adhering to the WHO IYCF recommendations is important for preventing malnutrition and poor growth during infancy, and ensuring good child health and development [3]. As a result, achieving optimal IYCF practices is a global priority [3].

In Ghana, breastfeeding and complementary feeding counselling are a major component of the nutritional intervention delivered at Child Welfare Clinics (CWCs) [4–6]. The 2022 Ghana Demographic and Health Survey [7] showed that sub-optimal IYCF practices were common, despite decades-long interventions by government and partners to improve them. For instance, among infants 0–5 months of age, 53% were fed exclusively, and 41% were fed a minimum dietary diversity of $\geq$ 5 of 8 food groups as of 2022 [7]. Additionally, studies on IYCF practices employing qualitative methods remain limited [8–12], with few exploring the facilitators and barriers influencing optimal IYCF practices [8, 12, 13]. Finally, not many studies have examined IYCF practices across varied geographic landscapes, which limits our understanding of how different environmental, cultural, and socio-economic factors influence IYCF practices in Ghana.

Ghana has a diverse geographic landscape (including various ecological zones ranging from the coastal plains in the south to the forest areas in the middle and the savannahs in the north) and a diverse population of various ethnic groups, each with its own cultural norms, traditions, and dietary preferences. These geographic and population factors may influence the feasibility of achieving optimal IYCF practices. Knowledge of the contextual facilitators and barriers to IYCF practices within the different districts may help develop targeted interventions and improve child nutrition in Ghana.

Between 2017 and 2019, the Ghana Health Service implemented the pilot Micronutrient Powder Initiative (MPI) in the Ho West, Tain, Talensi, and Tolon districts, in which multiple micronutrient powder (MNP) distribution was integrated into routine Child Welfare Clinic (CWC) services [14]. This study aimed to examine the IYCF practices among women in the geographically and ethnically diverse districts participating in the pilot program. The specific aims were to: (a) summarize the IYCF practices among the districts to identify differences and similarities, and (b) identify perceived motivators, facilitators and barriers to IYCF practices among the caregivers.

### Conceptual framework

The conceptual framework guiding this study was the theoretical framework from the Health Belief Model (HBM) [15], which we used to construct our key informant interview (KII) and focus group discussion (FGD) questions and to examine the participants' IYCF practices. The HBM comprises six key components: (i) perceived susceptibility (ii) perceived seriousness, (iii) perceived benefits of taking action (iv) perceived barriers to taking action, (v) cues to action, and (vi) self-efficacy. To operationalize these components we derived (a) *motivators* from perceived benefits and self-efficacy, e.g., positive outcomes mothers expected from adopting the IYCF recommendations and their confidence in their ability sustain the

adoption, (b) *facilitators* from the cues to action and modifying factors, including the external and internal prompts (e.g., access to resources, mother-to-mother support groups, and education from community health volunteers, etc.) that encourage individuals to adopt the IYCF recommendations, and (c) *barriers* from the obstacles mothers believed hindered their ability to adopt IYCF recommendations, including as financial constraints, belief that breastmilk alone was insufficient for the newborn child, lack of appropriate foods, and misinformation.

## Materials and methods

### Study setting

The study was carried out in the Ho West district in the Volta region, the Tain district in the Brong-Ahafo region (at the time the program began), the Talensi district in the Upper East region, and the Tolon district in the Northern region. These districts settings were described previously [14].

Briefly, the Ho West district located in the southern ecological zone had an estimated population of 82,886 [16] consisting predominantly of the *Ewe* ethnic group. Most inhabitants were crop farmers; the main staple foods included maize, plantain, cassava, and yam [17]. The Tain district in the Forest-Savannah transition zone within the middle ecological belt of Ghana had an estimated population of 115,568 [16] consisting predominantly of the *Banda* ethnic group. Most inhabitants were crop farmers and traders [18]; the main staple foods included corn, cassava, yam, and plantain. The Talensi district in the Guinea Savanna zone within the northern ecological belt had an estimated population of 87,021 consisting mainly of the *Talensi* ethnic group. Being a predominantly rural district, most inhabitants were crop farmers. The main staple foods were millet, sorghum, and yam. Finally, the Tolon district in the Guinea Savanna zone within the northern ecological belt had an estimated population of 118,101 consisting predominantly *Dagomba* ethnic group [16]. Most inhabitants were crop farmers; the main staple foods were millet, sorghum, maize, yam, and cowpea.

District-level data are not available, but in 2014 the prevalence of exclusive breastfeeding in the regions where the four districts are located ranged between about 54% in the Northern region and 69% in the Upper East region [19]. In addition, the proportion of children aged 6–23 months fed following all three recommended IYCF practices (food diversity, feeding frequency, and consumption of breast milk/milk) was about 11% in Brong Ahafo, 14% in the Northern region, 9% in Upper East, and 11% in Volta region [20].

### The Micronutrient Powder Initiative (MPI)

The Ghana Health Service (GHS) implemented the pilot Micronutrient Powder Initiative (MPI) with support from UNICEF, who supplied the MNPs and provided various logistical and operational support capabilities. Details of the MPI and lessons learned were published previously [14]. The program used various routine health service contacts, particularly the monthly Child Welfare Clinics (CWCs) to supply mothers and caregivers of children 6–23 months of age with micronutrient powders (MNPs) for home (point-of-use) fortification of complementary foods for those children, as an intervention strategy to combat anemia and micronutrient deficiencies [21].

Once enrolled, a child received 30 sachets of MNP (15 micronutrients per 1 g sachet) monthly for 3 months, followed by 3 months of not receiving any MNPs, and the cycle was then repeated until the child was 24 months of age. A child could stay in the program for a maximum of 18 months. Within communities, Community Health Nurses and other trained low-level health workers recruited from the communities where they live [22] monitored the MNP consumption under the Community-based Health Planning and Services (CHPS)

system [22, 23]. The previous study [14] showed that the MPI was well integrated into the GHS' institutional routine and that, the MNPs were generally well-accepted by mothers and caregivers.

## Study design and data collection

This was a qualitative study. As previously described [14], data were gathered through key informant interviews (KIIs) and focus group discussions (FGDs) conducted with mothers and caregivers who had a previous or current history of receiving MNPs for their infants according to CWC records. Within districts, we selected 6 sub-districts and then 2 mothers or caregivers per sub-district to participate in the individual KIIs, and up to 10 mothers or caregivers per sub-district to participate in the FGD. The GHS staff involved with the pilot program in each sub-district helped select mothers and caregivers who were 15–49 years of age (reproductive age) and were believed to have considerable knowledge about the MPI and the attitudes of women in the communities towards optimal IYCF.

Interview guides were used to collect information related to community practice of recommended IYCF guidelines, common beliefs around breastfeeding and infant care practices, IYCF recommendations perceived as hardest to follow, factors influencing the practice of IYCF recommendations, and IYCF support from around the community (See S1 File). All KIIs and FGDs were conducted in the local language. The KIIs and FGDs were conducted between 1$^{st}$ November 2019 and 20$^{th}$ December 2019. The interviews and discussions were audio-recorded and transcribed verbatim into English. All participants provided their informed written consent before they were interviewed.

Ethics approval for this study was obtained from the Ghana Health Service Ethics Review Committee (GHS-ERC 006/09/19). Additional information about the study design and data collection can be found in other studies [14].

## Reflexivity

Reflexivity is essential in qualitative research, and it involves how the researchers' experience, professional background and assumptions may influence the research process, including the data collection [24]. We recognize the influence of our professional backgrounds, experiences, and perspectives on the present investigation. Our collective expertise spans population studies, public health nutrition, and community-based nutrition assessments and interventions, with years of experience working in Ghana. Our backgrounds likely shaped our approach to examining the IYCF practices of the women in the pilot MPI, and it is possible that our professional roles may introduce biases, such as in data interpretation.

Throughout the study, however, we made a conscious effort to minimize biases by adhering to rigorous qualitative research methods and maintaining awareness of our assumptions and potential preconceptions. Having multiple investigators participate in the data analysis and interpretation helped ensure a balanced perspective, thereby mitigating potential individual biases. By incorporating quotes from the KIIs and FGDs, we aimed to present a holistic and authentic representation of IYCF practices in the districts. Our dedication to reflexivity guarantees the validity and reliability of our results, which help provide a more nuanced understanding of the facilitators and barriers to optimal IYCF practices in the four districts.

## Data analysis

We recorded the number of hours of audio recordings per district for the KIIs and the FGDs. All transcripts from the KIIs and FGDs were quality-checked and exported into QSR NVivo 10 (QSR International, Melbourne, Australia). These data were analyzed using a thematic

**Table 1. Summary statistics of the key informant interviews (KIIs) and focus group discussions (FGDs) with the mothers and caregivers, by district[1].**

|  | Ho West district | Tain district | Talensi district | Tolon district |
|---|---|---|---|---|
| Number of KIIs completed[1] | 12 | 12 | 12 | 12 |
| Number of FGDs completed[1] | 6 | 6 | 6 | 6 |
| Total number of FGD participants | 52 | 51 | 54 | 53 |
| Hours of KII audio recordings | 10.1 | 9.0 | 9.2 | 8.9 |
| Hours of FGD audio recordings | 8.5 | 7.5 | 7.8 | 7.8 |

[1]Mothers and caregivers had previous or current direct experience with feeding children with micronutrient power received; they were selected if they were believed to have considerable knowledge about the Micronutrients Powder Initiative (MPI) and the attitudes of women in the communities towards optima infant and young child feeding (IYCF) recommendations.

analysis approach [25]. All transcripts were read to understand the data and to identify emerging themes. All statements from transcripts relevant to the objectives of the study were assigned codes; similar codes were assigned the same themes. The themes emerging from the transcripts focused on infant and young child feeding practices including (a) level of mother's adherence to infant and young child feeding recommendations, (b) infant and young child feeding recommendations perceived as hardest to follow, and (c) perceived motivators, facilitators and barriers to IYCF practices. Findings on adherence to IYCF recommendations and IYCF recommendations hardest to follow were supported with mothers' quotes; findings on perceived motivators, facilitators and barriers to IYCF practices were summarized.

We implemented measures to enhance the rigour of the findings [26] as follows: first, three authors reviewed the codes and themes derived from the transcripts and discrepancies were resolved amicably. Second, all decisions made during the study design, data collection, and data analysis were documented. Third, findings from the KIIs were used to triangulate the findings of the FDGs.

## Results

We completed 12 KIIs and 6 FGDs in each district (Table 1). The 48 KII and 210 FGD participants were evenly or nearly evenly distributed across the 4 districts. The duration of the KII audio recordings ranged from 8.9 hours in Tolon to 10.1 hours in Ho West totaling 37.2 hours; that of the FGD recordings ranged from 7.5 hours in Tain to 8.5 hours in Ho West totaling 31.6 hours.

### Themes extracted from the FGDs and KIIs

Three themes emerged from the KIIs and FGDs, including (a) level of adherence to IYCF recommendations among mothers and caregivers, (b) IYCF recommendations perceived as the hardest to follow, and (c) perceived motivators, facilitators, and barriers to IYCF practices.

### IYCF practices among mothers and caregivers

**Early initiation of breastfeeding.**   Table 2 presents the summary of the KIIs and FGDs, which inquired about IYCF practices in each of the 4 districts. Across the four districts, participants observed that mothers generally initiated breastfeeding within one hour after birth, especially mothers who delivered at health facilities. A few quotes from the districts were:

*"As soon as you deliver in the hospital, after the mother and child have been attended to, within the first hour, the nurses tell you to give the breast to the child."* (KII, Ho West)

**Table 2. Similarities and differences of IYCF practices among mothers and caregivers participating in the Micronutrient Powder Initiative (MPI) in Ho West, Tain, Talensi, and Tolon districts.**

| IYCF practices | Ho West | Tain | Talensi | Tolon |
|---|---|---|---|---|
| *Early initiation of breastfeeding* | • Many mothers begin breastfeeding their children within one hour after birth, especially mothers who deliver at health facilities. | | | |
| *Exclusive breastfeeding for the first 6 months* | • Some mothers are able to exclusively breastfeed their children for the first 6 months. | • Many mothers exclusively breastfeed their children for the first 6 months. | | |
| *Timely introduction of complementary feeding* | • The timely introduction of complementary foods to children at 6 months of age in this district is unclear. | • The timely introduction of complementary foods to children at 6 months of age in this district is unclear. | • Many mothers begin complementary feeding on time. | |
| *Continuous breastfeeding for 2 years or beyond* | • Mothers breastfeed their children for a period between 18 and 24 months. | • Relatively few mothers continuously breastfeed their children for 2 years and beyond. | | • Many mothers breastfeed their children for up to 2 years. |
| *Give children 4 or more food groups per day* | • Many mothers/caregivers give their children 4 or more food groups since these foods are available on their farms and in the communities. | | • Many mothers are unable to feed their children with 4 or more food groups because of a lack of different food varieties. | • Many mothers/caregivers give their children 4 or more food groups since these foods are available on their farms and in the communities. |
| *Feeding children aged 6–8 months 2 times daily* | • Mothers commonly feed children 6–8 months of age at least twice daily in addition to breastfeeding. | | | |
| *Feeding children aged 9–23 months 3 times daily* | • Mothers/caregivers feed their children aged 9–23 months at least three times daily in addition to breastfeeding. | | • The number of times mothers feed their children depends on the availability of money. | • Mothers/caregivers feed their children aged 9–23 months at least three times daily in addition to breastfeeding. |
| *Active feeding of children during or after illness* | • Most mothers "coerce" their children who are ill or recovering from illness to eat so that they can recover speedily. | | | |
| | | • Mothers "pamper" their sick children to eat, to ensure children have enough energy to recuperate quickly. | | |
| *Complementary feeding recommendations hardest to follow* | • Giving children foods from 4 or more food groups due to lack of financial means. | | | • Timely introduction of complementary feeding since children aged six months find it difficult to eat any food apart from breastmilk. |

*"We breastfeed them because when you go to the hospital to give birth, the midwife will see to it that you start breastfeeding the child within the first 1 hour after giving birth."* (FGD, Tain)

*"What I know is that the child is breastfed right after childbirth. Immediately the child starts crying you have to breastfeed him/her."* (FGD, Talensi)

*"In this community when a woman delivers, whether at home or the health facility, she initiates breastfeeding within the first one hour."* (FGD, Tolon)

**Exclusive breastfeeding for the first 6 months.** In the Tain, Talensi and Tolon districts, the KIIs and FGDs revealed that more mothers tended to exclusively breastfeed their children for the first six months compared to mothers in the Ho West district. A few quotes from the districts were:

*"Not a lot of people are able to do it because they complain that the children suckle too much. Thus, they give them water and food"* (KII, Ho West).

*"We learn that the child should get to 6 months of age before we start giving him/her water and food. So, whether it's harmattan or rainy season, we try hard to do it."* (FGD, Tain)

*"We feed the children with breast milk only till the six months of age. Then afterwards, we give water and food."* (FGD, Talensi)

*"When you deliver, it is recommended to give only breast milk for six months. The child will not eat any food or drink water. After the six months, you start by giving "kooko". In this community we practice that."* (FGD, Tolon)

**Timely introduction of complementary feeding.** Regarding timely introduction of complementary feeding, mothers in the Talensi and Tolon districts demonstrated a strong adherence to that recommendation. In contrast, the level of adherence to that recommendation among the women in Tain and Ho West districts was unclear. Some of the quotes from the districts were:

*"That is what I was telling you that the child cannot eat the food because they have taken breastmilk for a longer period, so they are used to it. The breastmilk tastes sweeter than the food he/she is about to start consuming."* (KII, Ho West).

*"Some mothers go beyond 6 months before giving their children food, while others don't. In my case, I couldn't wait till 6 months before feeding my child."* (FGD, Tain)

*"We breastfeed for the whole six months. However, once the children get to six months of age, we start giving them other foods besides breastmilk."* (FGD, Talensi)

*"By six months of age, the breast milk is no longer sufficient for the child. So, you introduce other foods to the child to supplement the breast milk. It is not every food the child can eat at six months of age. There are specific foods that can be given."* (KII, Tolon)

**Continuous breastfeeding for 2 years or beyond.** Regarding continuous breastfeeding for 2 year or beyond, mothers in the Ho West and Tolon districts demonstrated a strong adherence, whereas mothers in the Tain and Talensi districts tended to indicate a low level of adherence. A few quotes from across the districts were:

*"Some mothers stop breastfeeding when the child is 2 years old. Exactly 2 years old or some few days to two years, they take them off the breast. In the community, a lot of mothers breastfeed their children for two years."* (KII, Ho West)

*"Over here, we are able to breastfeed our children to 2 years. We stop breastfeeding when the child is 2 years old to enable them to attend school. But those who are into white-collar jobs, need to resume work early so they wean their children at one and a half years."* (FGD, Ho West)

*"With my children, I am not able to breastfeed them till two years since they don't eat well. But with my current child, I breastfed him for 2 years."* (FGD, Tain)

*"I have 3 children. As for my children, I stopped breastfeeding one of them exactly 1 year and eight months, while I breastfed the other for 1 year and ten months. I do this because some of them are well-grown within these periods. However, other mothers stop breastfeeding when their children are 3 years old, while others also stop when their children are 2 years old. Every parent decides long to breastfeed their children."* (FGD, Talensi)

*"They are doing it. They are breastfeeding their children for up to two years. Some mothers even breastfeed their children for up to three years."* (KII, Tolon)

**Giving children 4 or more food groups per day.**   Mothers in the Ho West, Tain, Tolon districts were more likely to feed their children with 4 or more food groups, whereas in the Talensi district, this practice tended to be less common. Some of the quotes from across the districts were:

*"We vary the children's food. Over here, leafy vegetables are common and so there is no reason to say you need money before giving the child food and so we always try mixing their food for them."* (KII, Ho West)

*"Yes, do it: we use things like groundnuts, anchovies, fried maize, beans, powdered milk, margarine, etc. If you blend it into a powder (Tom brown), you can keep it and use it little by little for the child. It makes the child strong."* (FGD, Tain)

*"In a day, we give the children food from at least four food groups. For example, children consume tuo zaafi with vegetables. Then they eat fish and soya beans."* (FGD, Tolon)

**Feeding children aged 6–8 months 2 times daily.**   Feeding children aged 6–8 months 2 times daily in addition to breastfeeding was a common IYCF practice in the four districts. A few quotes from the districts were:

*"We give the children food in the morning and the evening, that is, at least breakfast and supper."* (KII, Ho West)

*"After breastfeeding, we give them porridge to drink then later in the day you give them something different. They eat up to 3 times in addition to the breast milk."* (KII, Talensi)

*"Up to eight months of age, if the child likes eating, you can give the 'kooko' [porridge] twice or three times in addition to breast milk. There are no specified times for the breast milk. Once you realize that the child is hungry you give the food for him to eat."* (KII, Tolon)

**Feeding children aged 9–23 months 3 times daily.**   Mothers in the Ho West, Tain, Tolon, and districts tended to feed their children aged 9–23 months at least three times daily in addition to breastfeeding. In the Talensi district, mothers observed challenges to adhering to this recommendation, citing low availability of funds in the households. Some of the women explained:

*"We feed the children three times a day with different foods like ewo, akple, leafy soup, herrings and salmon. So, we feed them about thrice a day."* (FGD, Ho West)

*"When there is money for food, I feed him/her 3 times a day but when there is no money, I feed him/her 2 times and then breastfeed him/her to make up for what he didn't eat."* (KII, Talensi)

*"The child who is 9 to 23 months old can be given food three to four times in a day in addition to the breast milk."* (FGD, Tolon)

**Active feeding of children during or after illness.** The participants observed that mothers "coerced" or "pampered" their children who were ill or recovering from illness to eat so that they could recuperate speedily. Some participants explained:

*"Some children have difficulty feeding when they're sick but the mothers have to 'force' them to eat on most occasions."* (KII, Tain).

*"When the child is ill, you pamper them in ways to make eating possible which would help in the healing process. Anything the child wants, you do it."* (FGD, Tain)

*"When a child is sick, he/she doesn't have an appetite for food so you'll have to lure him/her to eat. After he/she has eaten, he/she can regain his/her strength or at times you coerce him/her to eat. If you don't force the child, he/she won't eat the food."* (FGD, Talensi)

*"If the child is sick, he/she eats less compared with when he is well. In that case, you will have to force him/her to eat in small quantities."* (KII, Tolon)

**IYCF recommendations perceived as the hardest to follow.** The participants perceived some of the complementary feeding recommendations as the hardest to follow. Giving children food from 4 or more food groups per day was perceived by mothers and caregivers in Ho West, Tain, and Talensi as the most challenging IYCF recommendation. A few explanations were:

*"It isn't everyone that is financially sound to be able to buy all those foods for the child. The mother may not have money to buy different varieties of food for the child to eat, therefore she will force the child to eat 'banku' that she has prepared for herself. I think varying the menu of foods for the child is difficult for us."* (FGD, Ho West)

*"Those foods are not so common around here. What we have is corn, beans and groundnuts. With the groundnuts, it is mixed when the corn is milled; and also, you can get these leafy vegetables around here. For the other kinds of food to be prepared and eaten within specific intervals, it is difficult to come by around here."* (KII, Tain)

In the Tolon district, the timely introduction of complementary feeding was perceived as the hardest IYCF recommendation for mothers, citing the difficult for children aged 6 months it to eat any food apart from breast milk.

*"Because they always complain that when they give the child food, the child doesn't eat because of the exclusive breastfeeding. When they give the food, the child doesn't take it. The child only wants the breast milk"* (KII 1, Tolon).

*"Six months when you are beginning to give food to the child. With 'kooko' [porridge], the child finds it difficult to take because he/she is taking the breast milk and what you are now going to give to him/her now is different"* (KII 2, Tolon)

## Motivators, facilitators and barriers to IYCF practices

**Table 3** shows the motivators and facilitators of IYCF practices in the four districts; **Table 4** shows the barriers to IYCF practices. The IYCF practices include: (1) early initiation of breastfeeding, (2) exclusive breastfeeding for the first 6 months, (3) timely introduction of complementary feeding, (4) continuous breastfeeding for 2 years or beyond, (5) giving children 4 or

**Table 3. Motivators/facilitators of practicing IYCF guidelines among mothers and caregivers participating in the MPI in the Ho West, Tain, Talensi, and Tolon districts.**

| IYCF practices | Motivators[1]/Facilitators[2] | Ho West | Tain | Talensi | Tolon |
|---|---|---|---|---|---|
| *Early initiation of breastfeeding* | • Mother-to-mother support groups and community volunteers educate mothers on the need to initiate breastfeeding. | √ | √ | √ | √ |
| | • Midwives ensure that mothers initiate breastfeeding immediately after delivery. | √ | √ | √ | |
| | • Frontline nurses educate mothers on the importance of early initiative of breastfeeding during antenatal clinics and home visits. | √ | √ | √ | √ |
| | • Frontline nurses and community health volunteers inform mothers about the benefits of colostrum. | √ | √ | √ | √ |
| *Exclusive breastfeeding for the first 6 months* | • Midwives and nurses teach mothers how to breastfeed their children. | | √ | | |
| | • Frontline nurses educate mothers on the benefits and consequences associated with exclusive breastfeeding during antenatal clinics and child welfare clinics (CWCs). | √ | √ | √ | √ |
| | • Midwives educate mothers to practice exclusive breastfeeding after delivery. | | √ | | |
| | • Community health volunteers educate mothers on the need to practice exclusive breastfeeding. | | √ | | √ |
| | • Mothers are afraid to give food and water before 6 months since it will make their children ill. | √ | √ | | |
| | • Desire to adhere to the recommendations given by frontline nurses during antenatal clinics and child welfare clinics (CWCs). | | √ | √ | √ |
| | • A belief that exclusively breastfed children have improved wellbeing. | √ | √ | √ | √ |
| | • Mothers perceive exclusive breastfeeding as a natural family planning technique. | | √ | | |
| | • Some mothers do not go to the farm nor do household chores (such as fetching water and cooking) after delivery for some months. | √ | | | √ |
| | • Mother-to-mother support groups monitor mothers at home to ensure they are practising exclusive breastfeeding. | √ | | | |
| *Timely introduction of complementary feeding* | • A belief that children begin to feel for food at 6 months. | √ | | | |
| | • Frontline nurses educate mothers on the timely introduction of complementary feeding. | √ | | | |
| | • Frontline nurses inform mothers to start complementary feeding when they bring their children to CWC at 6 months. | | √ | √ | √ |
| | • A belief that breastmilk is not sufficient for children after 6 months. | | √ | | √ |
| | • Desire to adhere to the recommendations given by frontline nurses. | | | √ | |
| | • Mother cook separately for their children. | √ | | | |
| *Continuous breastfeeding for 2 years or beyond* | • Frontline nurses educate mothers to practice continuous breastfeeding for 2 years or beyond during CWCs. | | √ | | |
| | • A belief that a child is not eating well after 6 months of age. | | | √ | |
| | • Community health volunteers educate mothers to practice continuous breastfeeding for 2 years or beyond. | √ | | | |
| *Give children 4 or more food groups* | • Mother cook separately for their children. | √ | | | |
| | • Frontline nurses educate mothers on the various food groups to feed their children during CWCs and community durbars. | √ | √ | | √ |
| | Mothers feed children with fruits when fruits are in season. | | √ | | √ |
| | Children consume fruits, especially oranges and bananas since they are less expensive. | | | | √ |
| | Green leafy vegetables are common in villages/communities. | √ | √ | | |
| | Husbands/partners provide the finance to purchase different foodstuffs and also purchase foodstuffs from neighboring communities for the child's food preparation. | √ | √ | | |
| | • Some mothers vary their children's food daily. | √ | | | |
| *Feeding children aged 6–8 months 2 times daily* | • Mother-to-mother support groups monitor mothers to ensure they adhere to feeding their children 2 times daily. | √ | | | |
| | • Mothers routinely feed children twice daily, especially in the morning and afternoon. | √ | √ | | √ |
| | • Community volunteers educate mothers to feed their children twice daily in addition to breast milk. | | √ | | |
| | • Mothers routinely feed children at least 3 times daily. | | | √ | |

(*Continued*)

**Table 3.** (Continued)

| IYCF practices | Motivators[1]/Facilitators[2] | Ho West | Tain | Talensi | Tolon |
|---|---|---|---|---|---|
| *Feeding children aged 9–23 months 3 times daily* | • Mothers routinely feed children trice daily (in the morning, afternoon, and evening). | √ | √ | | √ |
| | • A belief that breastmilk is not sufficient for children. | | | √ | |
| *Active feeding of children during or after illness* | • Mothers cook their children's favorite foods to entice them to eat. | √ | √ | √ | |
| | • Mothers' habit of pampering sick children to eat. | √ | √ | √ | √ |
| | • Eating is necessary to allow children to take their medicines and to ensure speedy recovery. | √ | √ | √ | √ |

more food groups, (6) feeding children aged 6–8 months 2 times daily, (7) feeding children aged 9–23 months 3 times daily, and (8) active feeding of children during or after illness.

Across the eight IYCF practices, relatively more motivators/facilitators were identified compared to barriers. More motivators/facilitators were identified for exclusive breastfeeding for the first 6 months, particularly for the Talensi District:

> *"Please, we are taught at the weighing centre [child welfare clinics] that it helps make the children intelligent. So, having that in mind, it encourages us to breastfeed for exclusively six months, before introducing foods to them."* (FGD, Ho West)

> *"Most women do exclusive because we have been educated on the significance of exclusive breastfeeding, unlike the past."* (KII, Talensi)

> *"Health workers have instructed us to practice exclusive breastfeeding when we went to the hospital, so we practice it."* (FGD, Talensi)

Feeding children aged 9–23 months 3 times daily had the least motivators/facilitators. Regarding barriers to IYCF practices, exclusive breastfeeding for the first 6 months, and continuous breastfeeding for 2 years or beyond emerged as the common barriers experienced by mothers. In the Ho West district, mothers frequently faced various barriers to practicing exclusive breastfeeding for the first six months. In the Tain district, mothers frequently faced various barriers to maintaining continuous breastfeeding for two years or beyond. No barriers were identified for active feeding of children during or after illness. A few quotes from the participants were:

> *"There are times that some mothers do not have milk in the breast so it doesn't flow early. We sometimes report it to the nurse at the clinic and we are sometimes told to give the child food since he/she may be hungry."* (FGD 1, Ho West)

> *"Most times, the majority of the mothers are not very mature. They have their mother-in-law at home who controls them. The mother-in-law will inform them that "when I gave birth to my son, I gave him 'kooko' [porridge] when he was 2 months. My son is the one responsible for this pregnancy and you cannot be in my home and not feed the child with food.' So, they become scared of their mothers-in-law and do what they tell them to do against what health workers tell them to do. That's what we have observed in some of the homes."* (FGD 2, Ho West)

> *"Breastfeeding for a whole period of two years is the hardest. This is because mothers also want some ease. So, they take their children to school to concentrate on their work."* (KII 1, Tain)

**Table 4. Barriers to practising IYCF guidelines among mothers and caregivers participating in the MPI in the Ho West, Tain, Talensi, and Tolon districts.**

| IYCF practices | Barriers[3] | Ho West | Tain | Talensi | Tolon |
|---|---|---|---|---|---|
| *Early initiation of breastfeeding* | • Some mothers think they do not have sufficient breast milk to feed newborns after delivery. | √ | √ | | √ |
| | • When some mothers deliver at home, they do not initiate breastfeeding immediately after delivery. | | √ | | √ |
| | • When some mothers deliver at home, colostrum is not given to the newborns. | | | √ | |
| | • A belief that Caesarean delivery delays the early initiation of breastfeeding. | √ | | | |
| *Exclusive breastfeeding for the first 6 months* | • A belief that mothers may have insufficient breast milk force some mothers to feed their children with water and porridge. | √ | √ | | √ |
| | • Some mothers-in-law feed their grandchildren with water and porridge before 6 months. | √ | | √ | |
| | • Some nurses give water to children before 6 months of age. | √ | √ | | |
| | • Mothers give water and food to their children, especially when they cry since they think their children are thirsty and hungry. | √ | √ | √ | |
| | • Mothers give herbal concoctions to their children to treat and protect against certain illnesses. | | √ | | |
| | • Dry weather condition makes some mothers give water to their children. | √ | √ | | |
| | A belief that breastmilk alone was insufficient for the child. | | √ | | √ |
| | • Cultural practice of giving water to newborns immediately after delivery. | | | √ | √ |
| | • Some mothers give food because their children suck too much. | √ | | | |
| | • Some mothers do not have time to feed their children since they go to the farm and sometimes leave the children behind. | √ | | | |
| *Timely introduction of complementary feeding* | • A belief that when complementary feeding is "delayed" till 6 months of age, children then find it difficult to eat other foods besides breast milk. | √ | | √ | |
| *Continuous breastfeeding for 2 years or beyond* | • Some children wean themselves from breastfeeding before 2 years. | √ | | | |
| | • Need to return to school and work/resumption of work, e.g., students, apprentices, formal sector workers. | √ | √ | √ | |
| | • Some children start schooling before 2 years so their mothers wean them before starting school. | | | √ | |
| | • Some mothers wean their children before 2 years when they think those children are "well-grown". | | | √ | |
| | • Some mothers perceive that breastfeeding makes them grow lean/lose weight. | √ | √ | | |
| | • Some mothers perceive that the color of breast milk changes. | | √ | | |
| | • Some children do not eat well when they are breastfed till 2 years. | | √ | | |
| | • Some children suck breastmilk a lot since they do not get adequate food to eat. | √ | √ | | |
| | • Some mothers do not get adequate food to eat to breastfeed their children for 2 years. | | √ | | |
| | • Short birth interval among women, usually less than 2 years. | | | | √ |
| *Give children 4 or more food groups* | • Lack of financial means, since the appropriate foods are not always readily available. | | √ | | √ |
| | • Some mothers do not have time to cook and feed their children with 4 or more food groups. | √ | | | |
| | Meat as an animal source of food is uncommon in some communities. | √ | | | |
| | • Large family sizes with short birth intervals make some mothers concentrate/focus on the youngest child to the detriment of other younger children, especially those less than 2 years old. | | √ | | |
| | • Some foodstuffs are not available in some communities. | | | | √ |
| *Feeding children aged 6–8 months 2 times daily* | • Some mothers reportedly do not have time to feed their children twice daily. | √ | | | |
| *Feeding children aged 9–23 months 3 times daily* | • Mothers have "no time" to cook food for their children. | √ | √ | | |
| | • Some mothers with large family sizes are not able to feed their children 3 times daily. | | √ | | |
| | • Lack of financial means to provide food 3 times daily. | | | √ | |
| | • Scarcity of food in the community. | √ | | √ | |
| | • Some children breastfeed more rather than eat well. | √ | | | |

[1] Motivators were defined as reasons women/caregivers adhere to IYCF guidelines.

[2] Facilitators were defined as factors that promote women's/caregivers' ability to adhere to IYCF guidelines.

[3] Barriers were defined as factors that inhibited women's/caregivers' ability to adhere to IYCF guidelines.

*"Breastfeeding for 2 years is difficult because some mothers work, so once the child gets to a certain stage they stop and concentrate on their work even when the baby is not yet 2 years."* (FGD, Tain)

## Discussion

The purpose of this study was to examine the IYCF practices among mothers participating in the pilot MPI program in Ho West, Tain, Talensi and Tolon districts in Ghana. Findings from the key informant interviews (KIIs) and focus group discussions (FGDs) with mothers across the four geographically and ethnically diverse districts suggest that women generally adhered to the infant and young feeding practices, such as early initiation of breastfeeding, exclusive breastfeeding for first 6 months, give children 4 or more food groups, feeding children aged 6–8 months 2 times daily, and active feeding of children during or after illness, among others. Expanding the MPI program, which strengthens IYCF knowledge and practices among mothers and caregivers, to additional districts in Ghana could enhance the health and welfare of children under 5 years of age.

It was not surprising that mothers in the Tolon district demonstrated a strong adherence to the eight recommended IYCF practices, as those mothers had fewer barriers to IYCF practices. These findings highlight the necessity of bolstering health promotion programs in every district to improve mothers' and caregivers' adherence to IYCF practices.

Regarding early initiation of breastfeeding, mothers frequently had a positive perception of colostrum since they were educated by health workers about the benefits of colostrum. Education by health workers motivated mothers to feed their infants with colostrum. This finding supports studies in Ghana which found that mothers fed their infants with colostrum [27–31]. Education on the importance of colostrum for the wellbeing of infants should be strengthened by health workers during antenatal and postnatal clinics and home visits to help sustain the progress made in creating positive mothers' and caregivers' perceptions about the need to feed their infants with colostrum.

The study also found that giving children 4 or more food groups and timely introduction of complementary feeding were perceived as the most challenging recommendations to be followed by mothers. Giving children 4 or more food groups was perceived as the most challenging recommendation for women in Tain, Talensi and Ho West districts, due to lack of funds to purchase the different variety of foods. This finding is not surprising for two reasons. First, the incidence of multidimensional poverty in the regions (Brong Ahafo, Upper East, and Volta regions) of the three districts ranges between 49% and 68% which is higher than the national average of 46% as of 2017 [32]. Second, adequate dietary diversity among infants and young children is a major nutritional concern in the regions (Brong Ahafo, Upper East, and Volta regions) of the three districts. As of 2014, less than 29% of all children (breastfed and non-breastfed) aged 6–23 months in the three regions consumed food from 4 or more food groups [20]. Poor dietary diversity among children aged 6–23 months is associated with an increased risk of undernutrition and morbidity [3, 33].

The challenge of timely introducing complementary feeding to infants necessitates a reevaluation of health workers' education approach. Health education sessions should be more participatory, allowing for feedback and addressing mothers' and caregivers' unique challenges. Additionally, health workers should consistently reinforce education on timely introducing complementary feeding during postnatal clinics and home visits to empower mothers and caregivers to overcome their challenges with timely introducing complementary feeding.

Furthermore, social intervention programs, such as Livelihood Empowerment Against Poverty (LEAP), should be expanded to assist mothers and caregivers in providing their children

with food from 4 or more food groups. The LEAP is a cash transfer program that provides a bi-monthly stipend to vulnerable households, including households with pregnant women and mothers with infants. Among other things, LEAP aims to improve household food consumption, health and nutrition.

The study found that more mothers in the Tolon district reported that their children had difficulty consuming any food apart from breast milk at 6 months because the children were used to breast milk. This finding corroborates a study in England which found that mothers experienced difficulty feeding their children with complementary foods (lumpy solids) at 6 months [34]. This finding highlights the need for health workers in the Tolon district to educate mothers and caregivers on innovative strategies to employ when introducing complementary food to their infants to reduce their children's resistance when introduced to complementary foods.

The main motivators of IYCF practices identified include: frontline nurses and community health volunteers educated mothers about the benefits of optimal IYCF practices; a desire to improve the wellbeing of children; a desire to adhere to the recommendations given by frontline nurses, especially during antenatal clinics; routine feeding of children in the morning, afternoon, and evening by mothers; eating enable children to take their medicines and recuperate quickly; and exclusive breastfeeding being a natural family planning technique. These findings corroborate studies in Ghana, Ethiopia and West Africa which found that mothers who attended antenatal care were more likely to practice exclusive breastfeeding and appropriate complementary feeding practices [35–41]. In Nigeria, lactating mothers reported that the benefits of exclusive breastfeeding to children motivated them to practice exclusive breastfeeding [42]. Regarding exclusive breastfeeding as a family planning technique, a study by Adongo et al. [43] in Ghana also found that exclusive breastfeeding was perceived as a contraceptive method. Health practitioners should highlight the reasons why mothers and caregivers need to adhere to IYFC practices, as identified in this study, in their design of health education activities on IYCF to enhance mothers and caregivers' adherence.

The main facilitators of optimal IYCF practices were frontline nurses, community volunteers, midwives and mother-to-mother support groups educating mothers about the benefits of optimal IYCF practices; midwives and frontline nurses teaching mothers how to breastfeed; midwives ensuring mothers initiate breastfeeding immediately after delivery; and mothers cooking the favorite foods of their children and pampering them to eat, especially sick children. These findings corroborate other studies in Ghana, Nigeria, and West Africa which found that mothers who were aware of the benefit of exclusive breastfeeding to enhance their child's growth were more likely to practice it [36, 44, 45]. In Tanzania, Mgongo et al. [46] found that mothers practiced exclusive breastfeeding because it protects children against illness. Also, this study supported other studies in Ghana and South Africa that highlight the substantial role played by health workers in adopting optimal IYCF practices [28, 29, 38–40, 47]. Policymakers and health practitioners should strengthen the factors that promote mothers' and caregivers' adherence to IYCF guidelines, as highlighted in this study, in the development and execution of IYCF interventions to improve the health and wellbeing of children under five years of age.

The main barriers of IYCF practices identified include: insufficient breastmilk makes some mothers feed their children with water and porridge; some nurses and grandmothers (mothers-in-law) give water to children before 6 months; a perception that children are hungry and thirsty makes mothers give children water; the need of nursing mothers to return to work or school; some mothers not having time to cook and feed their children; mothers growing lean/ losing weight; giving herbal concoctions/medicine; and lack of financial means to purchase foods not readily available in community. These findings are similar to studies in Ghana, sub-

Saharan African countries, and Europe which found that grandmothers; losing weight; giving herbal medicine; children feeling thirty; insufficient breast milk; and resumption of work were barriers to the practice of breastfeeding and exclusive breastfeeding [13, 28, 29, 40–42, 46, 48–59]. In addition, studies in Ghana, Rwanda, and Kenya identified a lack of finance as a barrier to optimal infant and young child feeding [8, 60, 61].

In collaboration with relevant stakeholders (including the Ministry of Employment and Labor Relations, religious and traditional authorities, and assembly members), health professionals should address the factors that inhibit women's and caregivers' ability to adhere to IYCF guidelines. For instance, health professionals should collaborate with the Ministry of Employment and Labor Relations to start a national conversation on the need to expand the duration of maternity leave, which is at least 12 weeks, to help promote the practice of exclusive breastfeeding among mothers employed in the formal sector. Also, in partnership with traditional authorities and assembly members, health professionals should educate community members that breastmilk for the first 6 months is sufficient for their infants since it contains adequate food and water. Hence, there is no need to give food and water, including herbal concoctions/medicine, to infants less than 6 months.

A major strength of this study is that the KIIs and FGDs with participants were conducted in geographically and ethnically diverse districts and this provides a comprehensive knowledge on IYCF practices. A few weaknesses of the study deserve mention. First, participants for this study were purposively selected and therefore, the findings may not be generalized to all districts in Ghana. Second, participants were recruited with the help of Ghana Health Service staff at the sub-district level and this may be a source of bias. This may influence the study's findings and necessitates a cautious interpretation.

## Conclusions

This study demonstrated that mothers in the four districts generally adhered to the infant and young feeding recommendations, such as early initiation of breastfeeding, exclusive breastfeeding for the first 6 months, giving children 4 or more food groups, feeding children aged 6–8 months 2 times daily, and active feeding of children during or after illness, among others. However, mothers in the Tolon district adhered to the recommended IYCF practices more than mothers in the Tain, Talensi and Ho West districts. This emphasizes the necessity of enhancing education on IYCF practices to maintain the success that has been achieved.

Mothers in the Ho West district reported more barriers to IYCF practices, followed by mothers in the Tain, Talensi, and Tolon districts. We recommend that health practitioners, stakeholders and policymakers design targeted interventions to address context-specific barriers to improve IYCF practices in the various districts. Future studies should examine whether mothers' adherence to IYCF recommendations is related to children's nutritional status. This research would help ascertain whether adherence to IYCF recommendations translates into improved child nutritional status.

## Supporting information

**S1 File. Interview guides for data collection.**
(DOC)

## Acknowledgments

We thank the Director of the Family Health Division, GHS, Dr. Patrick Aboagye, for his support. We are also grateful to the Regional Directors of GHS for the Ho (Dr. Timothy Letsa),

Bono region (Dr. Kofi Issah), Upper East (Dr. Winifred Ofosu), and Northern (Dr. John Abenyeri) regions. We are also grateful to all the Ho West, Tain, Talensi, and Tolon district Nutrition Officers. We thank all the women who participated in the study.

## Author Contributions

**Conceptualization:** Seth Adu-Afarwuah.

**Data curation:** Frank Kyei-Arthur, Seth Adu-Afarwuah.

**Formal analysis:** Frank Kyei-Arthur, Seth Adu-Afarwuah.

**Funding acquisition:** Seth Adu-Afarwuah.

**Investigation:** Frank Kyei-Arthur.

**Project administration:** Frank Kyei-Arthur, Seth Adu-Afarwuah.

**Resources:** Jevaise Aballo, Abraham B. Mahama, Seth Adu-Afarwuah.

**Supervision:** Frank Kyei-Arthur, Seth Adu-Afarwuah.

**Validation:** Frank Kyei-Arthur, Seth Adu-Afarwuah.

**Writing – original draft:** Frank Kyei-Arthur, Seth Adu-Afarwuah.

**Writing – review & editing:** Frank Kyei-Arthur, Jevaise Aballo, Abraham B. Mahama, Seth Adu-Afarwuah.

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
