## [Decision Letter · Decision Letter 0]

9 Apr 2024

PONE-D-23-30376Infant and Young Child Feeding practices among mothers in the pilot Micronutrient Powder Initiative in four geographically and ethnically diverse districts in GhanaPLOS ONE

Dear Dr. Adu-Afarwuah,

Thank you for submitting your manuscript to PLOS ONE. After careful consideration, we feel that it has merit but does not fully meet PLOS ONE’s publication criteria as it currently stands. Therefore, we invite you to submit a revised version of the manuscript that addresses the points raised during the review process.

We look forward to receiving your revised manuscript.

Kind regards,

Veincent Christian Pepito

Academic Editor

PLOS ONE

 [This study was funded by a grant from the Government of Netherlands and UNICEF. UNICEF had a role in the decision to submit the results of this study.].  

[The first author is an academic editor for PLOS ONE.]. 

Additional Editor Comments:

Dear Authors, thanks for your manuscript. The manuscript has potential but has to be revised further to make it acceptable for publication. Here are my comments:

1. My main issue with the manuscript is the lack of conceptual framework that could guide the analysis. I do not know if there is any implicit conceptual framework that allowed you to categorize factors into motivators, facilitators, and barriers but this framework, whatever that is, should be explicitly mentioned. If you peruse COREQ or SRQR, this is one of the main requirements and this could make or break your paper.

2. There are many previous studies on the topic in Ghana. How is the current study addressing literature gaps or adding something new?

https://journals.sagepub.com/doi/10.1177/0379572117742298

https://cdn.who.int/media/docs/default-source/nutrition-and-food-safety/complementary-feeding/cf-guidelines/qualitative-review-preferences-equity-resources-acceptability-and-feasibility.pdf?sfvrsn=996ad2df_3

https://www.ncbi.nlm.nih.gov/pmc/articles/PMC10867453/

3. There is no such thing as a cross-sectional qualitative study. A qualitative study could either be a phenomenology, grounded theory, case study, or ethnography, among others. Please clarify.

4. The tables and narratives are good now but you are telling instead of showing. In presenting qualitative studies, I would prefer authors that show, instead of tell. In practice, this means that I would like you to show the specific vignettes in addition to the narrative and tables in the manuscript. The reason for this is I want to make my own conclusions about the vignette or dialogue instead of taking the perspective of the author solely.

5. One thing that we found to be associated with IYCF practice is the way IYCF intervention was introduced to the mothers. Have you considered this possibility as well? https://internationalbreastfeedingjournal.biomedcentral.com/articles/10.1186/s13006-021-00400-5

6. The Discussion and Conclusion statements are weak and generic. Beyond comparing your findings to previous studies, please discuss the meaning of your findings for practice and policy there in Ghana. What interventions can be scaled up? What interventions should be scaled down? What misconceptions are present? What barriers should be addressed? What motivators and facilitators should be incentivized? What context-specific interventions should be developed from your findings?

7. Please have the manuscript reviewed by a native English speaker or an editor.

8. Please also add a reflexivity statement describing each of your backgrounds, assumptions, beliefs, and work experiences that could affect the work being presented.

Please also address the comments of Reviewers 1 and 2.

Reviewers' comments:

Reviewer's Responses to Questions

**Comments to the Author**

1. Is the manuscript technically sound, and do the data support the conclusions?

Reviewer #1: Yes

Reviewer #2: Yes

2. Has the statistical analysis been performed appropriately and rigorously? 

Reviewer #1: Yes

Reviewer #2: Yes

3. Have the authors made all data underlying the findings in their manuscript fully available?

Reviewer #1: No

Reviewer #2: Yes

4. Is the manuscript presented in an intelligible fashion and written in standard English?

Reviewer #1: Yes

Reviewer #2: No

5. Review Comments to the Author

Reviewer #1: The topic of the study was noteworthy as it addressed one of the most important public health issues both at the study site and elsewhere. The authors have organized the manuscript well and written it in clear language. However, the following points must be addressed:

1. Describe the number of themes extracted from the study in the abstract and the result section in the manuscript.

2. Try to avoid quantifying adherence to the qualitative inquiry. It may be good to write as follows: Mothers in Tolo District reported good adherence to IYCF.

3. The study is about the participant’s perceived difficulties. Therefore, as much as possible, try to use words that represent qualitative expressions. For instance, instead of saying "hardest to practice,” use... was perceived as the hardest practice by the mothers.

4. It is better to start the result section with the sociodemographic and reproductive characteristics, followed by the number of themes extracted from the study.

5. Furthermore, each thematic area should be sequentially described. I suggest minor revisions to this part.

6. The number of women who supported the outcome (8 vs. 6) does not, in my opinion, differ statistically significantly enough to conclude that the practice was widespread. It is therefore recommended to utilize qualitative terminology to describe differences in practices.

With regards!

Reviewer #2: The manuscript is of important nature and the findings can be significant for gaining knowledge and producing more similar studies of longitudinal/causal relationships to improve nutritional and child care. However, there are several aspects of the document that need improving the descriptions plus the significance of findings. You will find several comments and suggestions in the attached file.

6. PLOS authors have the option to publish the peer review history of their article (what does this mean?). If published, this will include your full peer review and any attached files.

Reviewer #1: No

Reviewer #2: **Yes: **Dr. Alfredo L. Fort

---

## [Author Response · Author response to Decision Letter 0]

16 Jun 2024

Response to Reviewers

Editorial Office

Response: We have ensured that our manuscript meets PLOS ONE's style requirements, including those for file naming. 

Response: We thank the journal for the information. However, there are restrictions on sharing of these data since they belong to UNICEF. Interested persons can obtain the data from the corresponding author upon reasonable request and with permission from UNICEF. Dataset requests may be sent to Seth Adu-Afarwuah (sadu-afarwuah@ug.edu.gh).

Response: We have provided the grant number in the ‘Funding Information’ section and ensured the information in the ‘Funding Information’ and ‘Financial Disclosure’ sections match. 

 [This study was funded by a grant from the Government of Netherlands and UNICEF. UNICEF had a role in the decision to submit the results of this study.]. 

Response: We apologize for the error in the previous statement. We have corrected the financial disclosure statement as follows:

“Funding for this study was provided by UNICEF through an agreement with the University of Ghana (Contract No. 43210308). The funder had no role in study design, data collection and analysis, decision to publish, or preparation of the manuscript”. We have included the statement in our cover letter. 

5. Thank you for stating the following in the Competing Interests section: [The first author is an academic editor for PLOS ONE.]. 

Response: We have updated the statement in the Competing Interests section to “The first author is an academic editor for PLOS ONE. This does not alter our adherence to PLOS ONE policies on sharing data and materials”. We have included this statement in our cover letter. 

Response: We have included a caption for our Supporting Information files at the end of our manuscript and we have updated it in-text citations to match accordingly. See lines 759-760.

Editor 

1. My main issue with the manuscript is the lack of conceptual framework that could guide the analysis. I do not know if there is any implicit conceptual framework that allowed you to categorize factors into motivators, facilitators, and barriers but this framework, whatever that is, should be explicitly mentioned. If you peruse COREQ or SRQR, this is one of the main requirements and this could make or break your paper.

Response: We thank the Editor for this comment. We used the Health Belief

Model (HBM) described by Rosenstock et al. (1974) as our guiding conceptual framework to categorize factors influencing adherence to World Health Organization (WHO) Infant and Young Child Feeding (IYCF) recommendations into motivators, facilitators, and barriers. It was a major oversight that we did not explicitly include the framework in our initial submission. The HBM

is a widely accepted theoretical guideline for health behaviors in public health research [1]. In the revised manuscript, we have included a section on conceptual framework (lines 78-91) as follows:

“Conceptual framework

The conceptual framework guiding this study was the theoretical framework from the Health Belief Model (HBM) [2], which we used to construct our key informant interview (KII) and focus group discussion (FGD) questions and to examine the participants’ IYCF practices. The HBM comprises six key components: (i) perceived susceptibility (ii) perceived seriousness, (iii) perceived benefits of taking action (iv) perceived barriers to taking action, (v) cues to action, and (vi) self-efficacy. To operationalize these components we derived (a) motivators from perceived benefits and self-efficacy, e.g., positive outcomes mothers expected from adopting the IYCF recommendations and their confidence in their ability sustain the adoption, (b) facilitators from the cues to action and modifying factors, including the external and internal prompts (e.g., access to resources, mother-to-mother support groups, and education from community health volunteers, etc.) that encourage individuals to adopt the IYCF recommendations, and (c) barriers from the obstacles mothers believed hindered their ability to adopt IYCF recommendations, including as financial constraints, belief that breastmilk alone was insufficient for the newborn child, lack of appropriate foods, and misinformation.” 

2. There are many previous studies on the topic in Ghana. How is the current study addressing literature gaps or adding something new?

https://journals.sagepub.com/doi/10.1177/0379572117742298

https://cdn.who.int/media/docs/default-source/nutrition-and-food-safety/complementary-feeding/cf-guidelines/qualitative-review-preferences-equity-resources-acceptability-and-feasibility.pdf?sfvrsn=996ad2df_3

https://www.ncbi.nlm.nih.gov/pmc/articles/PMC10867453/

Response: In lines 59-63 of the introduction section of the revised manuscript, we have highlighted the gap in the literature, which this study sought to fill as follows:

“Additionally, studies on IYCF practices employing qualitative methods remain limited [3-7], with few exploring the facilitators and barriers influencing optimal IYCF practices [3, 7, 8]. Finally, not many studies have examined IYCF practices across varied geographic landscapes, which limits our understanding of how different environmental, cultural, and socio-economic factors influence IYCF practices in Ghana.”

3. There is no such thing as a cross-sectional qualitative study. A qualitative study could either be a phenomenology, grounded theory, case study, or ethnography, among others. Please clarify.

Response: The term “cross-sectional” is more commonly associated with quantitative studies, but it has been used to describe many qualitative studies as well [9, 10]. A quick search of “cross-sectional qualitative study" in Google scholar yielded “about 4,280 results”. We used "cross-sectional qualitative study" because our study captured a snapshot of participants' experiences and perceptions at a single point in time. In the revised manuscript, we have simply used “qualitative study” (Line 21 and Line 136).

4. The tables and narratives are good now but you are telling instead of showing. In presenting qualitative studies, I would prefer authors that show, instead of tell. In practice, this means that I would like you to show the specific vignettes in addition to the narrative and tables in the manuscript. The reason for this is I want to make my own conclusions about the vignette or dialogue instead of taking the perspective of the author solely.

Response: We have provided vignettes in the revised manuscript to support the narrative and tables. See pages 11-17, 20-22.

5. One thing that we found to be associated with IYCF practice is the way IYCF intervention was introduced to the mothers. Have you considered this possibility as well? https://internationalbreastfeedingjournal.biomedcentral.com/articles/10.1186/s13006-021-00400-5

Response: The study in the Philippines being referred to examined whether prenatal and postnatal peer counselor visits and membership in breastfeeding support groups were associated with early initiation of breastfeeding within 1 hour of birth and EBF at 6 months. The breastfeeding support groups were organized within the communities. While there was no significant association between peer counselor visits and early initiation or EBF at 6 months, membership in breastfeeding support groups significantly increased the odds of early initiation of breastfeeding by 1.49 times (95% CI 1.12, 1.98) and EBF by 1.65 times (95% CI 1.20, 2.24). 

As presented for our study in Ghana, we summarized IYCF practices and identified perceived motivators, facilitators, and barriers to IYCF practices among women from geographically and ethnically diverse districts who participated in the pilot Micronutrient Powder Initiative (MPI). Frontline nurses routinely conducted counselor visits, and breastfeeding support groups existed in the districts. Whether routine counselling visits and membership in breastfeeding support groups are associated with IYCF practices in these districts requires further investigation. 

6. The Discussion and Conclusion statements are weak and generic. Beyond comparing your findings to previous studies, please discuss the meaning of your findings for practice and policy there in Ghana. What interventions can be scaled up? What interventions should be scaled down? What misconceptions are present? What barriers should be addressed? What motivators and facilitators should be incentivized? What context-specific interventions should be developed from your findings?

Response: We have improved the discussion and conclusion sections of the revised manuscript. See pages 29-34, lines 436-565.

7. Please have the manuscript reviewed by a native English speaker or an editor.

Response: We have edited the manuscript to improve it. 

8. Please also add a reflexivity statement describing each of your backgrounds, assumptions, beliefs, and work experiences that could affect the work being presented.

Response: We have included reflexivity statements in the revised manuscript in lines 158-174 on page 8 as follows:

“Reflexivity 

Reflexivity is essential in qualitative research, and it involves how the researchers’ experience, professional background and assumptions may influence the research process, including the data collection [24]. We recognize the influence of our professional backgrounds, experiences, and perspectives on the present investigation. Our collective expertise spans population science, public health nutrition, and community-based nutrition assessments and interventions, with years of experience working in Ghana. Our backgrounds likely shaped our approach to examining the IYCF practices of the women in the pilot MPI, and it is possible that our professional roles may introduce biases, such as in data interpretation. 

Throughout the study, however, we made a conscious effort to minimize biases by adhering to rigorous qualitative research methods and maintaining awareness of our assumptions and potential preconceptions. Having multiple investigators participate in the data analysis and interpretation helped ensure a balanced perspective, thereby mitigating potential individual biases. By incorporating quotes from the KIIs and FGDs, we aimed to present a holistic and authentic representation of IYCF practices in the districts. Our dedication to reflexivity guarantees the validity and reliability of our results, which help provide a more nuanced understanding of the facilitators and barriers to optimal IYCF practices in the four districts.”

Reviewer #1

1. Describe the number of themes extracted from the study in the abstract and the result section in the manuscript.

Response: We thank the reviewer for the suggestion. The relevant section of the revised abstract (Lines 25-27) reads:

“Three emerging themes were: level of adherence to IYCF recommendations among mothers and caregivers; IYCF recommendations perceived as the hardest to follow; and perceived motivators, facilitators, and barriers to IYCF practices.”

Lines 211-214 of the results section reads:

“Themes extracted from the FGDs and KIIs

We identified three themes which emerged from the FGD and KIIs, including (a) level of adherence to IYCF recommendations among mothers and caregivers, (b) IYCF recommendations perceived as the hardest to follow, and (c) perceived motivators, facilitators, and barriers to IYCF practices.”

2. Try to avoid quantifying adherence to the qualitative inquiry. It may be good to write as follows: Mothers in Tolo District reported good adherence to IYCF.

Response: We thank the reviewer for the suggestion. In the revised paper, the relevant section of the abstract (Lines 28-32) reads: 

“Mothers in the Tolon district demonstrated adherence to IYCF practices, often citing the need for early initiation of breastfeeding, timely introduction of complementary feeding, and feeding children aged 9-23 months 3 times daily in addition to breastfeeding. In contrast, mothers in other districts faced challenges that hindered adherence.”

3. The study is about the participant’s perceived difficulties. Therefore, as much as possible, try to use words that represent qualitative expressions. For instance, instead of saying "hardest to practice,” use... was perceived as the hardest practice by the mothers.

Response: We have replaced the phrase “the hardest IYCF recommendations to be followed by mothers” with “perceived as the hardest to follow.” See lines 26-27, 213-214, 349.

4. It is better to start the result section with the sociodemographic and reproductive characteristics, followed by the number of themes extracted from the study.

Response: The revised manuscript includes socio-demographic characteristics of participants presented in Table 1.

5. Furthermore, each thematic area should be sequentially described. I suggest minor revisions to this part.

Response: In the revised manuscript, we have presented the results under each of the three themes sequentially. See lines 216-424.

6. The number of women who supported the outcome (8 vs. 6) does not, in my opinion, differ statistically significantly enough to conclude that the practice was widespread. It is therefore recommended to utilize qualitative terminology to describe differences in practices.

Response: We thank the reviewer for the suggestion. The revised manuscript incorporates qualitative terminology when describing differences in practices.

---

## [Decision Letter · Decision Letter 1]

16 Jul 2024

Infant and Young Child Feeding practices among mothers in the pilot Micronutrient Powder Initiative in four geographically and ethnically diverse districts in Ghana

PONE-D-23-30376R1

Dear Dr. Adu-Afarwuah,

We’re pleased to inform you that your manuscript has been judged scientifically suitable for publication and will be formally accepted for publication once it meets all outstanding technical requirements.

Kind regards,

Veincent Christian Pepito

Academic Editor

PLOS ONE

Additional Editor Comments (optional):

Reviewers' comments:

Reviewer's Responses to Questions

**Comments to the Author**

1. If the authors have adequately addressed your comments raised in a previous round of review and you feel that this manuscript is now acceptable for publication, you may indicate that here to bypass the “Comments to the Author” section, enter your conflict of interest statement in the “Confidential to Editor” section, and submit your "Accept" recommendation.

Reviewer #2: All comments have been addressed

2. Is the manuscript technically sound, and do the data support the conclusions?

Reviewer #2: Yes

3. Has the statistical analysis been performed appropriately and rigorously? 

Reviewer #2: N/A

4. Have the authors made all data underlying the findings in their manuscript fully available?

Reviewer #2: Yes

5. Is the manuscript presented in an intelligible fashion and written in standard English?

Reviewer #2: Yes

6. Review Comments to the Author

Reviewer #2: The authors have added several examples of comments made by respondents, which have added substance to the findings. They have also improved on the expressions and descriptions for more clarity to viewers, which is good. I have included a few additional suggestions in the attached file.

7. PLOS authors have the option to publish the peer review history of their article (what does this mean?). If published, this will include your full peer review and any attached files.

Reviewer #2: **Yes: **Dr. Alfredo L. Fort

---

## [Editor Report · Acceptance letter]

23 Jul 2024

PONE-D-23-30376R1 

PLOS ONE

Dear Dr. Adu-Afarwuah, 

I'm pleased to inform you that your manuscript has been deemed suitable for publication in PLOS ONE. Congratulations! Your manuscript is now being handed over to our production team.

Kind regards, 

on behalf of

Mr Veincent Christian Pepito 

Academic Editor

PLOS ONE